# Early-Stage Dissolution Kinetics of Silicate-Based Bioactive Glass under Dynamic Conditions: Critical Evaluation

**DOI:** 10.3390/ma14123384

**Published:** 2021-06-18

**Authors:** Dagmar Galusková, Hana Kaňková, Anna Švančárková, Dušan Galusek

**Affiliations:** 1Centre for Functional and Surface Functionalized Glass, Alexander Dubček University of Trenčín, Študentská 2, 91150 Trenčín, Slovakia; hana.kankova@tnuni.sk (H.K.); anna.svancarkova@tnuni.sk (A.Š.); dusan.galusek@tnuni.sk (D.G.); 2Faculty of Chemical and Food Technology STU, Radlinského 9, 812 37 Bratislava, Slovakia; 3Joint Glass Centre of the IIC SAS, TnUAD and FChPT STU, Študentská 2, 91150 Trenčín, Slovakia

**Keywords:** ion release, dissolution kinetics, flow test, bioactive glass

## Abstract

This manuscript presents a systematic and detailed study of ion release from 45S5 bioactive glass to develop a methodology to directly monitor dissolved ions in a simulated fluid via inductively coupled plasma optical emission spectrometry (ICP OES). For the kinetic study, two dynamic tests, an inline ICP test and a flow-through test, are performed with the same flow rate, temperature, pH, ionic strength of the solution, and sample surface to leaching solution volume ratio. The flow-through test allows for the measurement of an initial dissolution rate, as well the maximum amount of any species released from the surface of the glass. In addition, the data from the inline ICP test are obtained by immediate and direct monitoring of ions from the first minutes of contact of the glass with aqueous fluids with pH values of 4 and 7.4. The overall dissolution rates of the tested commercial bioactive glass in simulated body fluid (SBF) (pH 7.4) were significantly lower compared to the initial rate acquired. The methodology developed in this study can be applied to monitor the controlled release of ions with additional therapeutic functionalities, where the amount of ions released in the first minutes can be critical for the resulting biological performance.

## 1. Introduction

In the context of bioactivity, a specific biological response at the interface of a material is often desired. For instance, an ability to form hydroxyapatite (HA) is a commonly required and studied feature of materials used in bone-replacement applications. The evaluation of the ability of an implant material to form HA relies on the appropriate use of in vitro tests with suitable methods for diagnosing calcium phosphates, which are responsible for bridging osseous tissue with the material. For decades, the Kokubo recipe [1,2] has been routinely used in bioactivity and degradation testing with simulated body fluid (SBF) to identify materials with in vivo bone reactivity or tissue regeneration characteristics. The Kokubo recipe has also been cited in the ISO 22317 standard [3], which describes a method for detecting apatite formed at the surface of a material after submersion in a SBF. To prove the precipitation of apatite-like phases and the formation of interfaces attached to material, tests are designed to achieve oversaturation with respect to the desired phases and are often carried out under static conditions. The bioactivity of a tested material is proven upon the evaluation of X-ray diffraction patterns with a small incidence angle against the surface of the sample [3] and the degradation of bioactive material by measuring ion release after immersion in a SBF for one or more days. The ISO 22317 standard [3] recommends a soaking time of four weeks. Studies have relied on detecting apatite-forming characteristics based on XRD patterns, structural studies [4,5], surface observation [6], and analysis of the concentration of ions detected in SBF [7,8]. Today, the ability of a material to form crystalline hydroxyl carbonate apatite based on in vitro static tests in a SBF solution is commonly linked to its bioactivity.

Other papers [9,10,11,12,13,14,15,16,17,18] have attempted to progress further and discuss the kinetics and dissolution mechanisms of potential biomaterials. Scientific data on the topic have grown significantly in recent years, but it is difficult to compare the results from different studies due to discrepancies between the applied conditions. Some of these discrepancies include, but are not limited to, different specific surface areas, amounts of solution in contact with the material, and variations of the solutions with different ionic strengths [9,12,14,15] to simulate the human environment. Tests related to ion release under static conditions (closed system) prevail in the majority of published works, usually at different *S*/*V* ratios, i.e., the ratios between the surface area of the tested material, *S*, exposed to a specific volume, *V*, of the simulated body fluid. The sampling usually starts after one day of immersion. Macon et al. attempted to unify the testing conditions to allow for a comparison of the dissolution mechanisms and apatite nucleation [19]. A systematic methodology for interpreting data from tests to address the influence of various simulated body fluids on the kinetics and dissolution mechanism is also missing. Static tests predominantly rely on sampling at time intervals where the material in question is already in equilibrium with the fluid or is oversaturated with respect to the phase of precipitating from the solution. Unlike static conditions, dynamic tests are able to provide information about the early stages of dissolution. This is usually preferred when discussing the mechanisms of ion release from bioactive glasses.

A 4-component glass created 50 years ago by Hench, known as 45S5 Bioglass^®^ (trademarked by the University of Florida as a name for the original 45S5 composition), provided a starting point for numerous studies, experiments, and the invention of new glass compositions. This was also the starting point for all kinetic studies of bioactive glass in SBF. Much of the era was devoted to understanding the mechanism of the glass in terms of bonding with bone tissue, as well as the nature of the interface. What is already known about 45S5 glass has been summarized by Hench and Jones [20,21,22,23]. Besides the nature of the bioactive bond and the strength of the bond, they also answered questions related to the mechanisms involved in hydroxyapatite carbonate formation, where they described five surface reactions that are involved during the interaction of the Bioglass^®^ with a solution simulating the conditions in the human body. The mechanisms, which start with ion exchange [23], are already well-known and have been described, although Na^+^ released from the glass has been never experimentally measured in SBF due to the impossibility to quantitatively detect its contribution to the already high sodium concentration in simulated human plasma. The initial dissolution and release of Na has been described by molecular dynamics simulations of the interface between the 45S5 Bioglass^®^ surface and aqueous liquids [24]. Calcium and phosphorus release is usually measured after one day of exposure, and tests measuring ion release in hour-long intervals are mostly performed in Tris-buffered solutions or otherwise in solutions with varying concentrations of the major elements that are typical for a simulated plasma liquid [9,10,15,17,25,26]. 

Interpretation of experimental data obtained from immersion tests still needs to be clarified, especially when the dissolution rates are discussed and compared for systems with different chemical compositions and specific surface areas, as well as those performed in solutions with varying ionic strengths. The major interest of the scientific community has turned to the elemental quantification of the ions released after long-term exposure of the glass to SBF; however, during initial exposure, some ions can be rapidly released in bursts, resulting in concentrations that are detrimental or toxic [27,28,29,30] to human cells. Later, these ions can be trapped in complex compounds or phases precipitating from the fluid. Low concentrations of these ions determined later could lead to biased scientific conclusions. On the other hand, the identification of incorporated ions (other than Ca and P) and the extent of incorporation into the structure of any newly formed phase is difficult to determine and prove experimentally. Ions with additional functionalities (often called therapeutic ions) may also be added to glass in exaggerated amounts to achieve desirable biofunctionalities. Only a few authors have discussed the first hours of ion release under dynamic conditions in detail [9,15,16,30]; however, to the best of our knowledge, none of these results have been found in a simulated fluid. Monitoring the dissolution performance of a biomaterial from the first minute of its exposure to the SBF environment could provide insight regarding the chemical contribution to later precipitation reactions.

This study does not intend to jeopardize the methodologies of tests addressing the formation of apatite in SBF as a measure of bioactivity. The aim of this work is to highlight possible misinterpretations of data and address the kinetics and rates at which the particular ions are released from previously studied material systems. The 4-component glass created by Hench, known as Bioglass^®^, provided a starting point for numerous studies, and therefore it is used as a reference material for our kinetic studies. The commercially available glass (SCHOTT Vitryxx^®^ Bioactive Glass, SCHOTT AG, Landshut, Germany), with a composition identical to Hench’s Bioglass^®^ (24.5Na_2_O, 24.5CaO, 6P_2_O_5_, 45SiO_2_, in wt %), was selected and used to optimize and set the test conditions of the flow-through tests in the first minutes of the interaction between the material and fluid. This paper describes the possibility of theoretically interpreting dynamic test results of early stage dissolution. Specifically, it addresses the dissolution rates of potentially bioactive silicate-based glass systems. Quantitative information on calcium and phosphorus released into SBF from the immersed biomaterial, with precision of <3% RSD (relative standard deviation) and statistical significance distinguishable from the inherent concentrations in testing SBF, is another issue that is carefully evaluated from an analytical point of view. The ion release rates and kinetics of dissolution of these ions into other media frequently reported in the scientific literature (e.g., a solution buffered only with Tris and a solution with a pH of 4 to simulate an inflammatory process) are also discussed and critically evaluated here.

## 2. Experimental

### 2.1. Flow-Through (Dynamic) Test

An SBF buffer solution with a pH of 7.4 was prepared at 37 °C according to the protocol described by Kokubo et al. [1]. A reference solution prepared without the addition of characteristic simulated body ions (Ca, P, Mg, Na, Cl), buffered only with 0.05 M of Tris (SIGMA-ALDRICH, St. Louis, MO, USA) to maintain a pH of 7.4, was used in a comparative study for evaluating the kinetic characteristics of glass dissolution. Acetic acid (Analytika Ltd., Prague, Czech Republic) buffered with sodium acetate (SIGMA ALDRICH, St. Louis, USA) to a pH of ~4 [31,32,33] was also tested and labeled as Ac/NaAc. This solution is usually included in tests to simulate an inflammatory environment, which typically represents a local decrease of pH from 7 to 4.

An average weight of 225 ± 0.001 mg for the bioactive glass powder was determined based on Fagerlund’s [16] dynamic flow tests, where the concentration profile of ions was directly monitored by the optical emission spectrometer with inductively coupled plasma (ICP OES). In the flow-through tests, the aqueous solution continuously flowed through the sample placed in the corrosion cell with a volume of ~1 mL. The temperature was constant at 37 ± 0.1 °C and the flow rate was fixed at 0.56 mL/min to reflect the conditions in the human body and ensure ion quantification. Blood flow rates vary for various tissues. For instance, Cowles et al. [34] estimated that a 1 cm^3^ scaffold might experience a blood flow rate of about 70 mL/day in subcutaneous tissue; however, the rate used in this study was approximately ten times higher than that. This flow rate was used to provide the performed test, which was connected with the ICP OES instrument (further referred to as inline ICP test) in an inline configuration with robust conditions in plasma for detecting ions released from the glass.

The ion release performance in the flow-through tests was monitored in 20-min intervals for 6 h, and afterward in 1-day intervals during the following 4 days. To obtain the statistical variation of calcium and phosphorus determined in the SBF (matrix blank), the solution flowing through the reactor without the sample was also periodically sampled and analyzed at appropriate time intervals. The conditions of the flow-through tests were also applied for detecting the ion release performance of the glass at a pH of 4 in an aqueous solution buffered with sodium acetate.

### 2.2. Inline ICP Test 

The inline ICP test enabled monitoring of the first hour of glass-solution interactions in 2-min intervals. The whole test setup (the thermostatic bath, corrosion cell, peristaltic pump) was connected directly, i.e., inline, through tubing to the introduction system of the optical emission spectrometer (ICP OES, Agilent 5100 SVDV, Agilent Technologies Inc., Santa Clara, CA, USA). Parameters of the ICP OES applied to monitor the ion release are presented in Table 1. An internal standard of scandium (10 mg/L) was included in the inline sampling to deal with non-spectral interference. The limit of quantification for ions that are usually incorporated in various bioactive glasses but not dissolved in Kokubo designed SBF was determined by measuring 10 replicates of the matrix blank solution and was calculated as 10× sd, where “sd” is the standard deviation of the intensity of the signal of the respective element in the blank. The limit of quantification, with precision expressed as RSD%, was considered to be <5%. The ICP OES analysis was performed in an axial plasma viewing mode.

Calibration was performed by measuring certified reference materials (Analytika, Praha, Czech Republic) that were diluted to the required concentration range. Mixed calibration standards of calcium and sodium were prepared at 50, 100, 150, and 200 mg/L. Calibration solutions of phosphorus were prepared at 16, 32, and 65 mg/L, along with silicon at 1, 10, 20, and 50 mg/L. A possible shift of the slope for the calibration curve was monitored by remeasuring a control solution with dissolved ions of interest. Ion release of the studied glass was measured using the conditions of the inline ICP test from the beginning of the experiment in one-hour time intervals for solutions with pH values of 7.4 and 4.

### 2.3. Evaluation of ICP Data

The concentration of released ions in a time interval of 2 min was directly monitored using the ICP Expert software package. The total amount of leached elements (*nl_i_^t^)* was determined by the following equation [3]:(1)$$nl_{i}^{t}=(c_{i}\frac{F}{S}\Delta t)/x_{i}+nl_{i}{}^{t-\Delta t}\;(\mathrm{mg}/{\mathrm{cm}}^{2})$$
where *c_i_* is the concentration in mg/dm^3^ of a leached element *i* in time *t* (min), *F* is the flow rate (dm^3^/min, (*F* > 0), and *x_i_* is the weight fraction of the released element (*i*) in the glass. Variations in the particle shape, size distribution, and packing in the reaction cell can influence the availability of the effective surface area. Consequently, we included the surface area (*S*, cm^2^) of the glass powder, which was determined and calculated by the Brunauer-Emmett-Teller (BET) model.

In terms of the dissolution rate, *nl_i_* time dependences were considered for both the flow-through and inline ICP tests to obtain numerical information about the dissolution kinetics of bioactive glass. The inline ICP data were considered to determine the rate at the very early stage of dissolution. The intrinsic dissolution rate (*a_in_*), demonstrating the linear time dependence of ion transfer from glass to all tested solutions, was evaluated using a linear regression model with the use of the equation *nl_i_* = *a_in_ t*, where the kinetic parameter (*a_in_*) was calculated by the least squares method. The critical part of the inline ICP test evaluation was the selection of time intervals. The linear time dependence of ion release and the lack of back precipitation reactions were considered. The concentration of silicon was used as a dissolution tracer, since the element was not intentionally added to any fluid used for testing the bioactive glass and it provides information on the breaking of the silica network in the first hours of interaction with the fluid. The concentrations of calcium and phosphorus could be influenced by their partial precipitation at the glass surface.

A long-term four-day flow-through test was designed to demonstrate that the dissolution rate usually changes with the time and reaction progress, even under dynamic conditions. To compare the intrinsic rate *a_in_* calculated from the linear regression model for Si release from the inline ICP data, the output data from the flow-through tests were fitted with the semi-empirical equation described by Helebrant [35]. This semi-empirical equation was tested and discussed for the dissolution of various silicate-based glasses in aqueous solutions [35,36]. Fitting the time dependencies of the ions released into the solution *nl_i_ = f(t)* may be performed with Equation (2): (2)$$nl_{Si}=\frac{B_{Si}}{K_{Si}}\left[1-\exp\left(-K_{Si}t\right)\right]+W_{Si}t$$

By use of Equation (2), the empirical parameters *B*, *K*, and *W* were obtained. The dissolution rate of the glass matrix was calculated as the first derivative of *nl_Si_* (Equation (2)) at the dissolution time close to zero and is expressed by Equation (3): (3)$${\left(dnl_{Si}/dt\right)}_{t\to 0}=B_{Si}+W_{Si}$$

Equation (3) is comparable to the kinetic parameter (*a_in_*) calculated by the least squares method when using the inline ICP data. Bioactive glass is highly reactive and the overall rate of dissolution (*a*) could differ significantly from the rate determined in the early stage of dissolution, as discussed in [35]. Equation (4) considers the possible back-precipitation of released ions: (4)$$a=\frac{1}{\left(1-k^{-}\right)}{\left(\frac{dnl_{Si}}{dt}\right)}_{t\to 0}$$
where *k^−^* represents a back-precipitation parameter calculated from Equation (5) as the ratio between the leaching rate of the tracer (Si in this case) and the precipitating element *i* as t → ∞ (dimensionless):(5)$$k^{-}=1-\frac{{\left(\frac{dnl_{Si}}{dt}\right)}_{t\to \infty }}{{\left(\frac{dnl_{i}}{dt}\right)}_{t\to \infty }}$$
where ${\left(dnl_{Si}/dt\right)}_{t\to \infty }$ is given by the value *W_Si_*. In the case of silicate glasses, sodium or potassium is usually considered for calculating ${\left(dnl_{i}/dt\right)}_{t\to \infty }$. Due to the high content of sodium in the leaching solutions used here, the sodium release data feature high uncertainty. Consequently, the concentration profiles were evaluated using calcium as the ion (*i*), and these were acquired from the fitted (Equation (2)) flow-through experimental data *nl_Ca_ = f(t)*.

Calculations of interdiffusion coefficients were omitted based on the conclusion of Fagerlund [16], where a shift in the reaction control mechanism from diffusion-controlled toward surface-controlled can be expected for flow rates higher than 0.4 mL/min.

### 2.4. Dissolution of Selected Bioactive Systems

The commercially available SCHOTT Vitryxx bioactive glass powder (SCHOTT AG, Landshut, Germany) with d50 was further labelled as VB (4.0 ± 1.0 µm) and d95 was ≤20 µm. A chemical composition of 45SiO_2_-24.5CaO-24.5Na_2_O-6P_2_O_5_ (on a wt % basis) was used for the optimization of the experimental parameters for inline ICP measurement. It was also used for evaluating the kinetic performance of the glass by fitting the flow-through experimental data with the Helebrant empirical model shown in Equation (2) [35]. All elements, except sodium, that leached from the tested glass were monitored during the flow dynamics tests for all proposed solutions. Sodium release cannot be reliably determined due to its high concentration (>3 g/L) in SBF, and the same applies for Ac/NaAc solutions. After each test, the powder samples were washed with isopropyl alcohol and the surfaces were examined with a scanning electron microscope (SEM) (JEOL JSM-7600F, Tokyo, Japan).

### 2.5. Specific Surface Area

Nitrogen adsorption/desorption isotherms were measured (ASAP2020, Micromeritics ASAP 2020 Plus Physisorption, Norcross, GA, USA) at a temperature of −196 °C. Before the nitrogen adsorption/desorption measurements, each sample was degassed at 100 °C for 3 h. The BET model was applied to determine the specific surface areas of the samples. The pore size distribution was calculated through the DFT method (density functional theory) using the non-local density functional theory (NLDFT) kernel of equilibrium isotherms (desorption branch).

## 3. Results

With respect to bioactivity and the dissolution performance of bioactive glasses, the most discussed elements are calcium and phosphorus. These elements are also present in significantly high concentrations in fluids used to simulate human plasma. The theoretical concentrations of Ca and P vary depending on the procedure and recipe recommendations [9,13,37]. The accuracy of the measured concentrations depends on the robustness of the instrumental techniques, the effects of noise, errors, the concentration ranges of calibration curves, the linearity of calibration standards, blank contamination [38], and operator knowledge and skills. The SBF solution (labeled as matrix blank) was measured before each inline ICP test by analyzing the solution flowing through the reactor without the sample for at least 30 min. Figure 1 and Figure 2 summarize the concentration distributions of calcium and phosphorus in the SBF (matrix blank) from the inline ICP experiments. The calcium and phosphorus concentrations in SBF, considering a precision of <1 RSD%, are in the range of 92 to 100 mg/L and 28 to 32 mg/L, respectively (Table 2). The scattering of data for both elements needs to be carefully considered if bioactivity performance is discussed on the basis of ion release tests. As such, in this study, the matrix blank was always measured before the inline ICP test with the bioactive glass samples. The calculated mean values were subtracted from the concentrations of the ions originating from the interactions of tested bioactive glass with the SBF.

Only the measured values of the Ca and P concentrations outside the intervals shown in Table 2 were considered for evaluation. The concentrations of Ca and P above the maximal values indicated in Table 2 were considered as the amounts of released ions that originated from the dissolution process. The concentrations of Ca and P which fell to a reliability interval for the measurement indicate that the solid SBF system is in equilibrium and is considered to have zero contribution in terms of the interpretation of ion release with time, where *nl_i_ = f(t)*. The concentrations of Ca and P below the minimum values specified for both elements in Table 2 usually indicate the precipitation of some secondary alteration products from the SBF. The precipitation of apatite-like phases is predominantly considered here.

### 3.1. Evaluation of the Experimental Data from the Inline ICP Tests

To interpret the experimental data, we first evaluated the commonly expressed concentration profiles and discussed them in terms of *c_i_ = f(t)* (Figure 3) [9,12,39], where *c_i_* is the concentration of the ion (*i*) in mg/L as released over time (*t*). Second, we plotted the data such that *nl_i_ = f(t)* (Equation (1)) from both the inline ICP tests, i.e., for the inline data that were monitored with an optical spectrometer (Figure 4), and the classical flow-through (Figure 5) test, i.e., the data obtained from sampling at given time intervals. The results from the inline ICP test for Ca, P, and Si released from the VB glass during the first 60 min indicate that calcium was released at a higher rate than silicon. The concentration of phosphorus decreased slightly in the first 5 min of testing and then remained stable over time. This can be seen via observation of the graphical concentration trends. 

Subtraction of matrix blank for any particular element (*i*) and calculation of the converted amount of the element (*nl_i_*) according to Equation (1) was carried out for both the flow-through and inline ICP test data. The total surface area of the VB glass powder was 0.274 m^2^ when calculated by the BET model. This was included in the calculation of the released ion amounts. The experimental data from the inline ICP test show the kinetic performance for the first few minutes when the VB glass interacted with the leaching solution (Figure 4). Calcium and silicon were released to the SBF at the same rate during the first 15 min of the test. After that, a gradual decrease in the leaching rate was found. The *a_in_* values of the released ions, summarized in Table 3, were calculated for time intervals between 2 and 15 min. Calcium and silicon were released at approximately the same rate in the SBF, while phosphorus was depleted from the solution at a rate that was estimated to be 100-fold lower (−0.0529 ± 0.0003, ng·cm^−2^ s^−1^). At the very beginning of the test, a decline in the magnesium concentration in the SBF from 38 ± 0.3 to 32 mg/L was also detected. The kinetic assumptions based on evaluation of the time–concentration dependence (*c_i_ = f(t)*) differ significantly from those estimated from the time dependence (*nl_i_ = f(t)*).

The leaching rates of the ions released from the tested glass in the early stage of dissolution differed significantly depending on the leaching solution that was used. Even in the solutions with a neutral pH (SBF and Tris, pH of ~7) the initial release rates of all ions differed substantially. In SBF, calcium and silicon were released at approximately the same rate (Figure 4), while in Tris, the release rates of phosphorus and silicon were almost identical (Table 3). The measured rate of calcium release in the Tris solution significantly differed from the rates measured for Si and P. It is important to note that deionized water buffered with Tris is frequently used instead of SBF in studies addressing the kinetic behavior of bioactive glass in order to avoid problems with the saturation and precipitation of leached elements, which consequently affect the reliability of results. The highest release rates of Ca and P were measured during the tests in acetic acid buffered with sodium acetate. Unlike that in the Tris solution, silicon was released from the VB glass at approximately the same rate for both the SBF (pH 7.4) and Ac/NaAc solution (pH 4), where pH was measured at 22 °C (Figure 5). The results indicate that the ionic strength of a solution may be the key factor influencing the dissolution rate in the initial period of dissolution and not the pH. 

### 3.2. Evaluation of the Experimental Data Gathered from the Flow-through Tests

The conditions and the dynamic regimes were identical for both the inline ICP and flow-through tests. To validate the initial rates determined from the inline ICP tests, the rates for the early stage dissolution were also determined from the data obtained from the complementary flow-through tests over a course of four days. Plotting the *nl_i_ = f(t)* data resulting from the flow-through dynamic test in the SBF (Figure 6) demonstrated that the system approached equilibrium gradually within 24 h of the test.

After the first day, calcium was depleted from the solution in higher amounts than silicon. On the contrary, during the first day of the test, phosphorus leached from the VB glass at a level of <5 mg/L. For longer leaching times, its concentration was always below the minimal concentration of P in the SBF matrix blank. 

The rate of initial silicon release was estimated from the flow-through experimental data fitted with Equation (2) and was calculated from the fitted parameters (*B_Si_, W_Si_*) as ${\left(dnl_{Si}/dt\right)}_{t\to 0}$ = *B_Si_*+*W_Si_* (3) for a time approaching zero (0.613 ± 0.008, ng·cm^−2^·s^−1^). The results are comparable to the rate determined from the linear plot of the inline ICP test data calculated by the least squares method (0.546 ± 0.004, ng·cm^−2^·s^−1^, Table 3). Empirical parameters determined from the fitted flow-through *nl_i_ = f(t)* data are shown in Table 4. The same procedure was applied for determining the kinetic parameters of the glass tested in an aqueous solution buffered with sodium acetate to a pH of 4. The estimated initial leaching rates in the solution with a pH of 4, which were determined independently and were based on data from both the inline ICP and flow-through tests, were also comparable, achieving values of 0.59 ± 0.05 ng·cm^−2^·s^−1^ and 0.63 ± 0.04 ng cm^−2^ s^−1^, respectively. 

The overall rate of dissolution can differ because the estimation, which is based on the first derivative for *t→0*, does not account for the possible depletion of ions from the solution. Significant ion leaching from a material with low durability, such as bioactive glass, may disturb the equilibrium of the system and initiate the nucleation of new species. Steady states are also achieved under dynamic conditions when the rate of release of leachable ions remains constant, i.e., all available ions are leached from the surface layer or are bound in re-precipitated phases [35]. Independent of the pH of the solution, the rate of ion release gradually decelerates after a day of continuous flow of the solution through VB glass (Figure 7 and Figure 8). Calcium release decelerated considerably more than the release of silicon in our results. The saturation index for the precipitation of apatite phases was more than likely achieved during the SBF tests, so the initial rate of dissolution was expected to differ significantly from the overall dissolution rate. A summary of all parameters acquired from the fitted experimental flow-through data used to estimate the overall rate (*a*) is shown in Table 5.

Referring to the calculations (Equation (4)), the reference VB glass dissolved in the SBF with a pH of 7.4 at a rate (*a*) of ~0.2 ng·cm^−2^·s^−1^, which is 3 times slower than the initial rate of dissolution of 0.613 ng·cm^−2^·s^−1^ that was determined for the time interval between 0 and 15 min. The rate estimated for *t→0* from *nl_Si_* and the overall rate of the VB glass dissolving in the Ac/NaAc solution at a pH of 4 did not differ significantly, with values of ~0.62 and ~0.56 ng·cm^−2^·s^−1^, respectively.

Figure 9 shows micrograph images of glass grains before and after the leaching tests. Unlike the glass before testing (Figure 9A), electron microscopy examination revealed the presence of secondary products originating from back-precipitation reactions during the flow-through experiments (Figure 9B) in the SBF. Some intermediate phases of an unknown origin were also detected on the surfaces of glass particles after just 1 h of exposure to the Ac/NaAc solution with a pH of 4 during the inline ICP test (Figure 10B); however, the examination of the glass grains washed continuously with the same solution for 4 days (Figure 10A) did not show any evidence of precipitated phases enveloping glass grains, and the morphology resembled that of the glass grains before the tests (Figure 9A). The amounts of alteration products observed via electron microscopy were too low to be detected by XRD, and the identification of precipitated phases was thus not possible. 

## 4. Discussion

There is no uniform method for selecting media that simulate human plasma conditions to suitably study the kinetics of potentially bioactive materials, especially when the controlled release of ions with additional functionalities is considered. Unlike Kokubo’s SBF, which is used to study the precipitation of carbonate phosphate phases that form a firm interface with a potentially bioactive material, there are no standardized guidelines on which solution to use for kinetic studies in simulated human conditions. A solution without dissolved inorganic ions that is buffered to a pH of 7.4 with Tris is usually selected to stimulate the dissolution of bioactive materials under the conditions of the highest solubility and the lowest saturation with respect to calcium phosphate precipitation [10,11,25,40]. Acetic acid buffered with sodium acetate is used when an inflammatory environment is tested in vitro [15,17,40]. Other solutions with or without calcium or phosphates at various concentrations are also used [32,33]. Fluids containing proteins are of major interest for bioresponse investigations [11,18,25,41]. Data from static tests in SBF are often used to evaluate the kinetics of ion release, and the interpretations are frequently based on time–concentration trends starting from the first day of the experiment. At this point, there is already a high probability that a steady state has been achieved and the rate of dissolution of bioactive glass will gradually decelerate to zero, especially in solutions with a high ionic strength.

In this kinetic study, we acquired ion release data using dynamic flow-through testing, i.e., inline ICP testing (Figure 4, maximum duration of 1 h) and a flow-through test (Figure 6, maximum duration of 4 days) under identical conditions. Both test setups used the same flow rate, temperature, pH, ionic strength of the solution, and surface to volume ratio, and the ICP OES analysis was performed with the same range of calibration solutions. Although the concentrations of ions released from the glass in the flow-through tests were relatively low, they were still able to be reliably detected by the method and were distinguishable from the SB matrix blank. The solutions that were selected in this study are most commonly applied in testing the ion release of bioactive glasses, though these tests are mostly performed under static conditions. The release/exchange of ions is accompanied by a pH increase in the case of glass dissolution, which is an indicator of cation replacement in glass by protons provided by the solution [15]. The pH of the SBF solution immediately increased to 8 after placing the glass powder in the reaction cell, resulting in accelerated dissolution kinetics. The breaking of O–Si–O bonds might occur especially quickly due to the attack of OH^-^ ions which are available in an abundance in solutions with an increased pH of ~8. After the initial pH change here, the pH remained stable during the first hour of the flow-through test and then gradually decreased to 7.7. 

The starting pH (4) of the sodium acetate-buffered solution increased by about 35% during the first hour, where a 10% increase from the initial value was then detected after 4 h. Although Na^+^ ions could not be determined due to their already high concentrations in both solutions, it is likely that the concentrations are rapidly released with Ca^2+^ and are replaced by the H_3_O^+^ ions that exist in the surrounding environment [24,41]. The initial rates of Si release were comparable for all studied solutions in a range of 0.2 to 0.5 ng cm^−2^ s^−1^. It is generally acknowledged that basic aqueous solutions favor rupturing of the silica network in silicate glasses. Nonetheless, the same initial rate of Si dissolution was determined in this study for the Ac/NaAc solution, whose pH value increased to 5.4 after 10 min of exposure to the VB glass.

The studied 45S5 glass has a low durability, i.e., it is easily dissolved in an aqueous environment. It can be assumed that the local oversaturation with respect to Si was minimized based on the linear concentration and time dependence data for dissolved silicon in the initial time interval of 2 to 15 min (64 data points). This occurs due to the continuous flow of an aqueous solution over the dissolving material, irrespective of the used leaching medium.

The initial dissolution rates of the VB glass, expressed by the dissolution of Si, were found to be comparable for both the inline ICP and flow-through test and were close to ~0.6 ng·cm^−2^·s^−1^. If the conclusions on the mechanism of dissolution were based on the measured concentration–time dependencies, one would inevitably come to an incorrect conclusion. In all tested solutions, calcium was detected in higher quantities than silicon (Figure 3); however, speaking in terms of the release rates, silicon was released into the SBF at the same rate as calcium, signaling that the glass dissolved congruently here (Figure 4). Phosphorus in the form of dissolved PO_4_^3-^ ions could either react with magnesium in the SBF or be immediately consumed by some back-precipitation reactions (e.g., the formation of Posner’s clusters [42]). Depletion of the P ion was assessed to be a hundred times slower than the rate of release of Ca ions. No Ca, Mg, or P was present in the Tris solution at a pH of 7.4, and the initial dissolution rates of all elements (including phosphorus) were comparable. 

In all previously published works where conclusions have been based exclusively on the results of static tests, the formation of apatite phases is a predominant concern when assessing bioactivity; however, when addressing the dissolution behavior of a material through the kinetics of its dissolution, e.g., in bioactive glasses doped with therapeutic ions, flow-through tests should not be avoided. Moreover, static tests only provide information on the concentration of species once a steady state is achieved, i.e., the dissolution rates of ions released from bioactive glass are close to zero [35,36]. A steady state can be also be achieved under dynamic test conditions when the release rate of moveable ions remains constant; however, there is still an important question to be answered: Does any precipitation occur within the first few minutes of glass dissolution? A quantitative mathematic model for the kinetic and thermodynamic properties of octacalcium phosphate (OCP) and hydroxyapatite formation, developed by Carino et al. [41], suggests that in the first 100 min of dissolution, saturation is not achieved and no solid is formed. Our estimations for the initial dissolution rates, assessed from the first 15-minute time interval by the linear fitting of data from the inline ICP tests, excluded the formation of new phases in the first few minutes of monitoring ion release. This methodology could be applied to monitor therapeutic ion release and can be verified by additional flow-through tests for determining the initial rate of dissolution of bioactive silica-based glass materials under conditions for simulating human plasma. Since the conditions for saturation, with respect to the formation of new phases, are favorable in a SBF, static tests are more suitable for the assessment of bioactivity than for evaluating the initial ion release of a bioactive material. 

The behaviors of silicate bioactive glasses in aqueous solutions after long exposure times have been studied and described by many authors [19,20,23]. The transient formation of metastable phases exhibiting a fast formation rate is also possible, despite conditions that are thermodynamically favorable to apatite formation [42]. Among them, amorphous calcium phosphate (ACP) and octacalcium phosphate (OCP, triclinic) are two potential candidates [41] that have already been suggested as precursors of in vivo bone apatite formation [39,42]. Both compounds can be hydrolyzed to apatite in a subsequent step, and hydrated silica, along with a polycrystalline hydroxyl-carbonate apatite (HCA) bi-layer, is formed on the glass surface within ~10 h of contact with the SBF [23]. In this study, precipitates with different morphologies were observed on the glass grains after exposure to SBF or the Ac/NaAc solution with a pH of 4 (Figure 9B and Figure 10B). The attempt to identify the precipitated phases was unsuccessful. The amount of the given precipitated phase was below the detection limit of the X-ray powder diffraction. Based on visual observation of the plate-like morphology, we cannot draw any conclusions about the natures of the phases. Other calcium phosphates may also exhibit anisotropic or plate-like morphologies, e.g., monetite (CaHPO_4_) or brushite (CaHPO_4_⋅2H_2_O). Triclinic OCP also exhibits a characteristic petal-like morphology [42]. Nevertheless, our prime interest was to determine the overall rate (*a*) of dissolution for VB glass in both SBF and an Ac/NaAc solution with a pH of 4. The overall rate of glass dissolution (*a*) in SBF of ~0.2 ng·cm^−2^·s^−1^, estimated from Equation (4), differed significantly from the initial rate of ~0.6 ng·cm^−2^·s^−1^. Possible back-precipitation reactions involved in a later stage of dissolution could bias the information related to the real rates at which the ions are released into human fluids under in vitro conditions. Both the SBF and Tris solutions, which had an identical pH of 7.4, differed in their ionic strengths. Despite having the same pH, the rates at which ions were released into the solutions differed significantly (Table 3). As such, the ionic strength of a solution should also be considered when setting the conditions of testing in this regard. The Ac/NaAc solution differed not only in terms of the ionic strength and the buffer applied, but also in terms of the pH. These differences must be considered when setting the conditions for testing and when critically evaluating data which estimate ion release rates from bioactive glass during in vitro tests for simulating an inflammatory environment. When the appropriate methodology and data evaluation methods are applied, the kinetic parameters can be determined for any studied aqueous solution; however, a proper simulation of the actual conditions in the human body, to which all developed bioglass systems should be exposed, remains a challenge.

## 5. Conclusions

This manuscript presents the possibility to theoretically interpret dynamic leaching tests in the early stages of dissolution. The SCHOTT Vitryxx commercial bioactive glass, which is compositionally identical to Hench’s Bioglass^®^, was used in the experiments detailed here. The results are useful for those interested in estimating the dissolution rates of candidate bioactive glass systems. For the kinetic study, two dynamic tests, i.e., an inline ICP and flow-through test, were performed with the same flow rate, temperature, pH, ionic strength of the solution, and surface area to volume ratio. The ICP OES analysis was performed here using the same range of calibration solutions to acquire ion release data. The flow-through test allowed for the measurement of an initial dissolution rate, as well as the maximum amount of any species released from the surface of the glass. The inline ICP test yielded information on the kinetics of specific ion release in the first minutes of material exposure to aqueous solutions. This could differ significantly from the kinetics at the latter stages. The results revealed significant differences in the initial dissolution rates of the VB glass in various solutions with different pH and ionic strength properties. The tests were performed with powders here. Bulk samples can also be used, and the only limit in that regard is the volume/dimension of the corrosion cell. Different aqueous solutions and temperatures may be used. Since ion release is monitored directly in an inline ICP test, the limit of quantification for a particular ion in the solution as measured by ICP OES is the limitation of the methodology here. The methodology that was developed and verified in this study can be applied to monitor the controlled release of ions with additional therapeutic functionalities, where the amounts of released ions in the first minutes can be critical to the biological performance.

## Figures and Tables

**Figure 1 materials-14-03384-f001:**
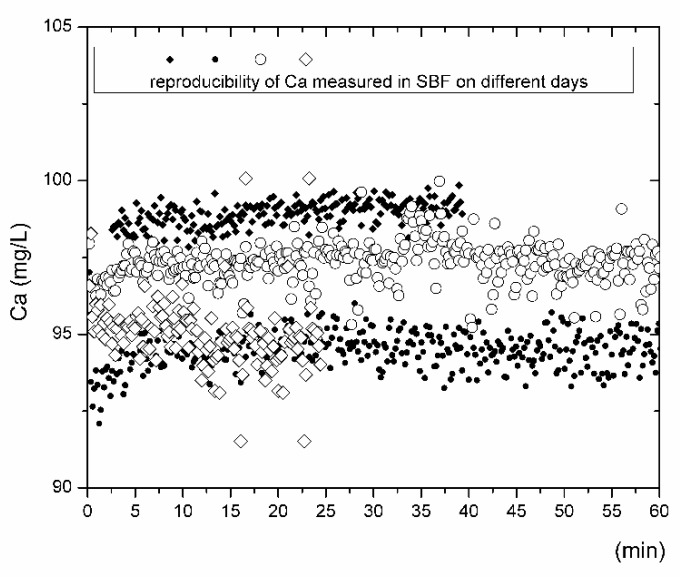
Reproducibility of the determination of Ca content during the inline ICP tests as measured on different days (represented by different symbols) in the fluid freshly prepared according to the Kokubo recipe [2].

**Figure 2 materials-14-03384-f002:**
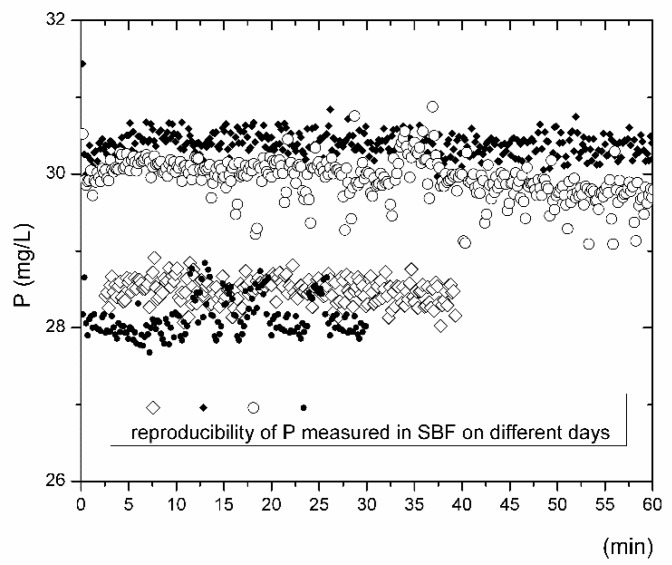
Reproducibility of the determination of P content during the inline ICP tests as measured on different days (represented by different symbols) in the fluid freshly prepared according to the Kokubo recipe [2].

**Figure 3 materials-14-03384-f003:**
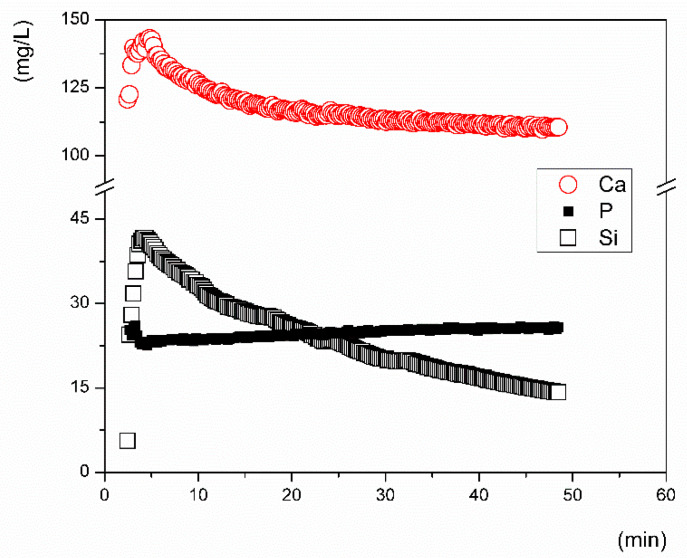
Inline ICP test results for ion release from the Vitryxx bioactive glass (VB) in the SBF solution with results expressed in terms of the weight concentration, i.e., mg of ion per liter of solution.

**Figure 4 materials-14-03384-f004:**
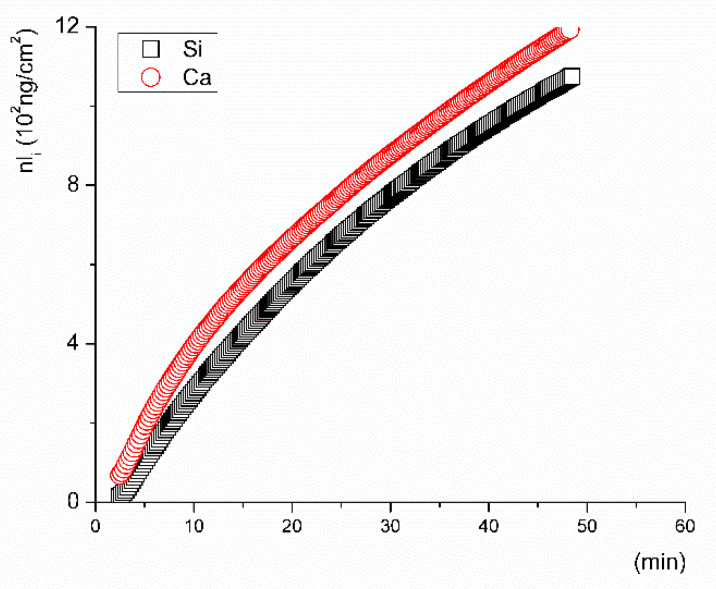
Inline ICP test results expressed as the cumulative converted ion (Si-squares and Ca-circles) amount from the VB glass in the SBF solution when normalized to its surface area (S).

**Figure 5 materials-14-03384-f005:**
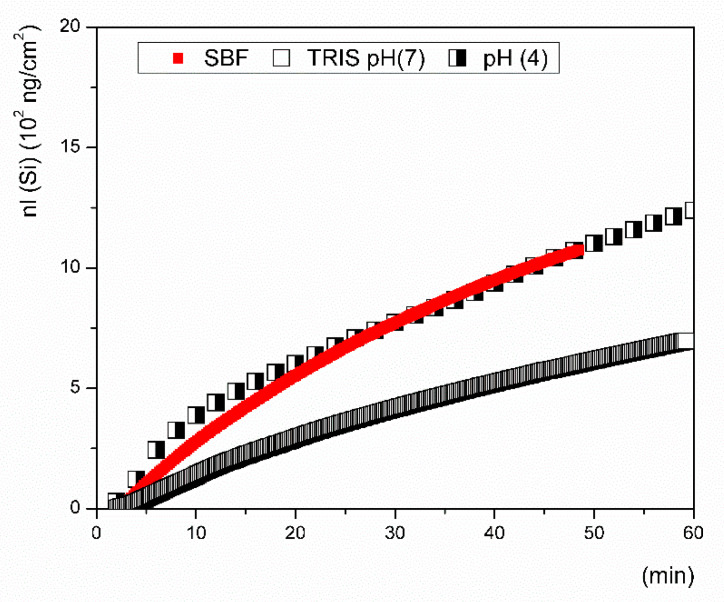
Ion release of silicon from the Vitryxx glass in deionized water buffered with Tris (empty squares), SBF (red squares), and the pH of the solution (pH of 4, half-filled squares) as monitored during the inline ICP tests.

**Figure 6 materials-14-03384-f006:**
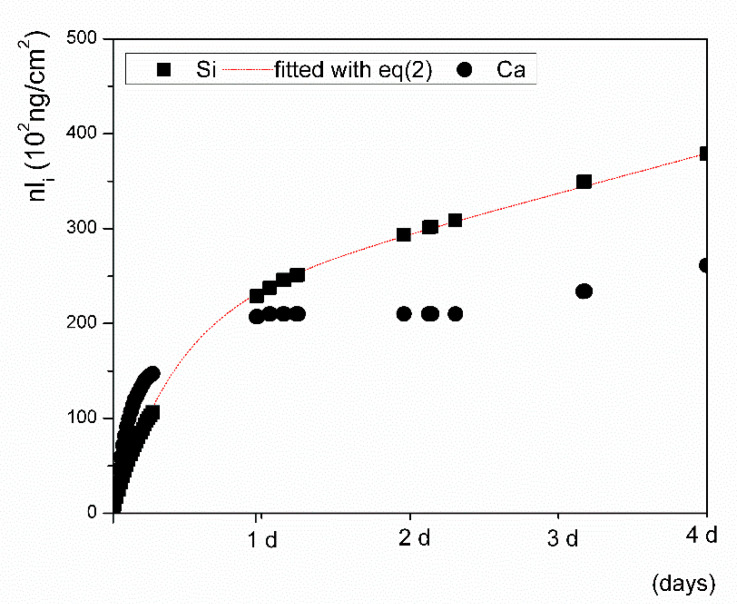
Flow-through test results of ion release in SBF for *nl(Si)* = *f(t)* (black square) as fitted with an empirical equation (dotted line) when compared to *nl(Ca)* = *f(t)*.

**Figure 7 materials-14-03384-f007:**
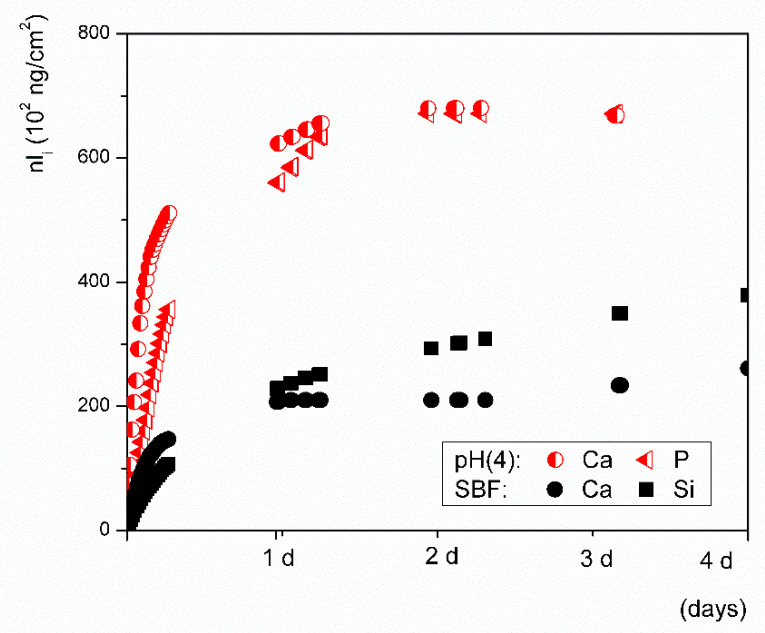
Ion release performance of Vitryxx glass in an aqueous solution at a pH of 4 when buffered with Ac/NaAc, nl(P and Ca) = f(t) (red triangles and circles), when compared to SBF nl(Ca and Si) = f(t) (black circles and squares).

**Figure 8 materials-14-03384-f008:**
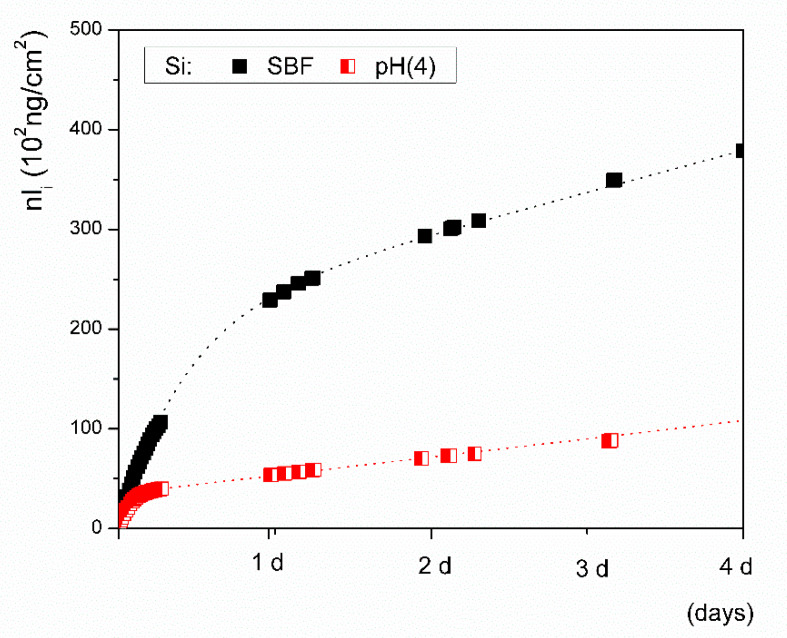
Silicon released from Vitryxx glass in the SBF (black squares) and in an aqueous solution at a pH of 4 (red squares). Lines represent the fit of nl(Si) = f(t) with empirical Equation (2).

**Figure 9 materials-14-03384-f009:**
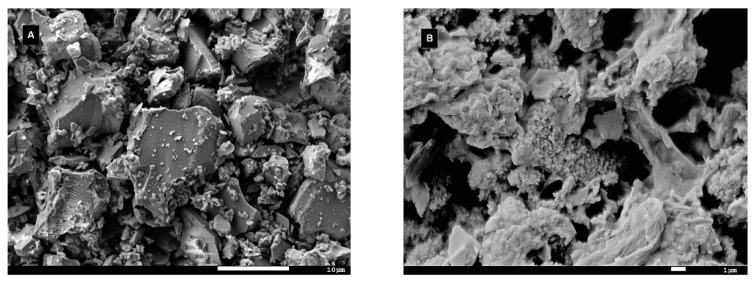
SEM micrographs (**A**) of VB glass powder before (**B**) testing and after the flow-through test in the SBF at a pH of 7.4.

**Figure 10 materials-14-03384-f010:**
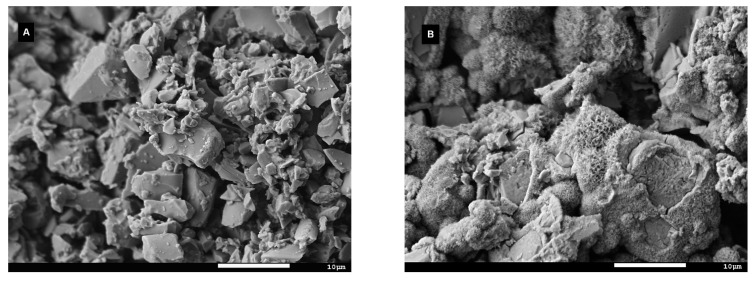
SEM micrographs (**A**) of the VB glass powder after the 4-day flow-through test (**B**) and the inline ICP test in Ac/NaAc fluid with a pH of 4.

**Table 1 materials-14-03384-t001:** Operating conditions of the inductively coupled plasma optical emission spectrometry (ICP OES) analysis in an axial viewing mode.

**RF Power (kW):**	1.2	**Background Correction:**	Fixed Point
**Carrier gas (mL/min):**	0.55	**Wavelength (nm):**	Ca (317.933), P (214.914), Si (288.158)
**Plasma gas (mL/min):**	12	**Read time (s):**	5
**Auxiliary gas (mL/min):**	1	**Number of replicates:**	3

**Table 2 materials-14-03384-t002:** The basic characteristics of the datasets obtained via the measurement of calcium and phosphorus with the ICP OES method.

Element	* N	Mean	sd	Minimum	Median	Maximum
		(mg/dm^3^)		(mg/dm^3^)		
Ca	1089	97	2	92	97	100
P	1097	29.7	0.8	28	30	32

* Total number of data points.

**Table 3 materials-14-03384-t003:** Parameters for estimating the initial ion release rates (*a_in_*) obtained from fitting the inline ICP data via linear regression *nl_i_ = f(t)* in an interval from 2 to 15 min as determined for all tested solutions.

*a_in_* ± sd	Ca	R^2^	P	R^2^	Si	R^2^
	(ng cm^−2^ s ^−1^)		(ng·cm^−2^·s^−1^)		(ng·cm^−2^·s^−1^)	
SBF, n = 64	0.619 ± 0.009	0.9886	−0.0529 ± 0.0003	0.9985	0.546 ± 0.004	0.9965
Tris, n = 64	0.427 ± 0.005	0.9919	0.294 ± 0.004	0.9907	0.299 ± 0.002	0.9984
Ac/NaAc, n = 32	3.5 ± 0.1	0.9888	1.79 ± 0.03	0.9978	0.59 ± 0.05	0.9872

*n* denotes the number of fitted points for data.

**Table 4 materials-14-03384-t004:** Empirical parameters with RSD% ≤ 2 as determined from fitting the *nl(Si)* flow-through data with Equation (2) when estimated for dissolution in SBF (pH of 7.4) and Ac/NaAc (pH of 4).

Flow-Through Test: *nl(Si) = f(t)*	R^2^	*B*	*K*	*W*
		ng·cm^−2^·s^−1^		
SBF	0.9993	0.566 ± 0.008	0.0027 ± 0.0001	0.047 ± 0.001
Ac/NaAc	0.9939	0.609 ± 0.038	0.017 ± 0.001	0.020 ± 0.001

**Table 5 materials-14-03384-t005:** Summary of parameters for the calculation of the overall dissolution rates (*a*) of the tested glass in selected fluids.

Figure 2	K^−^	${\left(dnl_{Si}/dt\right)}_{t\to 0}$	${\left(dnl_{Si}/dt\right)}_{t\to \infty }$	${\left(dnl_{Ca}/dt\right)}_{t\to \infty }$	a
		ng·cm^−2^·s^−1^			
SBF (pH of 7.4)	−1.6	0.613	0.047	0.0178	0.23
Ac/NaAc (pH of 4)	−0.12	0.629	0.020	0.0181	0.56

## Data Availability

The data presented in this study are available on request from the authors.

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
