# Peer review of "Early-Stage Dissolution Kinetics of Silicate-Based Bioactive Glass under Dynamic Conditions: Critical Evaluation"

_materials, 2021, doi:10.3390/ma14123384_

Round 1
Reviewer 1 Report
This study compares two methods, namely flow-through test and inline ICP test, to monitor the release of elements like Ca, Si, and P from a special bioactive glass during leaching with simulated body fluids or low pH acetate buffers. Overall, it is a well written manuscript and communicates the project justification, choice of materials and methods, and the final results and conclusions in an effective manner. Only minor changes are required, which are listed below, but otherwise this manuscript is suitable for publication.
Suggested minor edits:
- The abstract is not presented well. It does not explain that two types of methods are being compared and that one method was found to provide more details (about the initial 2 to 15 min) than the other. The conclusion did a better job of conveying this information than the abstract. In fact the term "inline ICP" is not even mentioned in the abstract. It may have been intentional, but it only confuses a reader. Therefore, please simplify the abstract and be as direct and specific as possible.
- Figures 1 and 2: The legend for the symbols on the charts are not provided. Please check and revise.
- There are two figure 9, which must be a mistake. Please edit the caption of Figure 10.
- Also, the captions for figures 9 and 10 should mention the test conditions such as the pH, whether it was a flow through test or an inline ICP test, and whether it was SBF or Ac/NaAc solutions.
Author Response
The authors would like to express their thanks to the Reviewer for insightful comments, which would help improving the quality of the manuscript. We did our best to respond to all reviewer comments and modify the manuscript accordingly. Detailed description of all changes made are attached in separate pdf document. The changes in the manuscript are highlighted through "Track changes" function. A native English speaker revised the language of the manuscript.

Reviewer 2 Report
The manuscript entitled ″ Early-stage kinetics of dissolution of silicate-based bioactive glass under dynamic conditions: Critical evaluation″ describes a methodology to access the ion dissolution kinetics from the bioactive glasses under dynamic conditions. The study is carefully planned and executed. The scientific conclusions are well supported by an adequate amount of data.
However, authors are required to provide the following details in the manuscript.
- In the flow-through (dynamic) test, why a 20 min interval was used to study the ion release performance in the initial 6 hrs. Is there a scientific reason behind this time selection?
- Did the authors convert powdered bioactive glass into a disc for flow-through dynamic test and inline-icp test?
- As the dissolution kinetics of bioactive glasses depend on chemical composition, crystallinity, surface roughness, solution composition, pH, and temperature. Can the methodology developed in this study be extrapolated to evaluate dissolution kinetics of bioactive glass with different surface roughness, crystallinity, and under different experimental conditions such as solution composition and temperature? A brief explanation pertaining to this could be added in the discussion section.
Author Response
The authors would like to express their thanks to the Reviewer for his/her insightful comments, which would help improving the quality of the manuscript. We did our best to respond to all reviewer comments and modify the manuscript accordingly. Detailed description of all changes made are in attached pdf document. The changes in the manuscript are highlighted through "Track changes" function. A native English speaker revised the language of the manuscript.

Reviewer 3 Report
the manuscript is very confusing and poorly written. the sentences are very long and puzzling with many linguistic and grammatical mistakes. the research totally is insignificant and does not provide any impact in the field. Not recommended to publish in this journal.
Author Response
The authors would like to express their thanks to the reviewer for her/his opinion. However, because objections expressed in the review were not specific, we were unable to modify the manuscript accordingly. A native English speaker revised the language of the manuscript.
Round 2
Reviewer 3 Report
The Authors have tried to improve the manuscript, however, there are still some flaws detected in the text which should be eliminated. the introduction is still unclear and confusing. the authors should explain what is the main concept of the research. some parts of the text are unnecessary and should be deleted. in the experimental part, the description of SEM analysis is missing.
Author Response
Please see attachment to see the authors response to the Reviewer 3 comments in 2nd round revision.
